# Exploring the Nutrition Strategies Employed by Ultra-Endurance Athletes to Alleviate Exercise-Induced Gastrointestinal Symptoms—A Systematic Review

**DOI:** 10.3390/nu15204330

**Published:** 2023-10-11

**Authors:** Tansy Ryan, Ed Daly, Lisa Ryan

**Affiliations:** Department of Sport, Exercise and Nutrition, School of Science and Computing, Atlantic Technological University, H91 T8NW Galway, Ireland; tansy.ryan@atu.ie (T.R.); ed.daly@atu.ie (E.D.)

**Keywords:** ultra-endurance, exercise, gastrointestinal, nutrition, intervention, symptoms

## Abstract

(1) Background: Participation in ultra-endurance sports, particularly ultra-running, has increased over the previous three decades. These are accompanied by high energetic demands, which may be further exacerbated by extreme environmental conditions. Preparation is long-term, comprising of sufficient exercise management, supportive dietary habits, and nutritional intakes for optimal adaptations. Gastrointestinal symptoms are often cited as causing underperformance and incompletion of events. Though the majority do not pose serious long-term health risks, they may still arise. It has been suggested that the nutritional interventions employed by such athletes prior to, during, and after exercise have the potential to alter symptom incidence, severity, and duration. A summary of such interventions does not yet exist, making it difficult for relevant personnel to develop recommendations that simultaneously improve athletic performance by attenuating gastrointestinal symptoms. The aim of this research is to systematically review the literature investigating the effects of a nutrition intervention on ultra-endurance athletes exercise-induced gastrointestinal symptom incidence, severity, or duration. (2) Methods: A systematic review of the literature was conducted (PubMed, CINAHL, Web of Science, and Sports Discus) in January 2023 to investigate the effects of various nutrition interventions on ultra-endurance athletes’ (regardless of irritable bowel syndrome diagnosis) exercise-induced gastrointestinal symptoms. Variations of key words such as “ultra-endurance”, “gastrointestinal”, and “nutrition” were searched. The risk of bias in each paper was assessed using the ADA quality criteria checklist. (3) Results: Of the seven eligible studies, one was a single field-based case study, while the majority employed a crossover intervention design. A total of *n* = 105 participants (*n* = 50 male; *n* = 55 female) were included in this review. Practicing a diet low in short-chain, poorly absorbed carbohydrates, known as fermentable oligosaccharides, disaccharides, monosaccharides, and polyols (FODMAPs), as well as employing repetitive gut challenges of carbohydrates, remain the most promising of strategies for exercise-induced gastrointestinal symptom management. (4) Conclusion: Avoiding high-FODMAP foods and practicing repetitive gut challenges are promising methods to manage gastrointestinal symptoms. However, sample sizes are often small and lack supportive power calculations.

## 1. Introduction

While some researchers recognize the term ultra-endurance activity as bouts of exercise that exceed four hours in duration, it has also been proposed that, on the basis of physiological and psychological stress, ultra-endurance activity must last for at least six hours [1,2,3]. Moreover, events may last multiple days or weeks, with athletes oftentimes competing for the best course completion time [4]. Ultra-endurance activity may refer to running, cycling, swimming, skiing, or multisport training [3]. During races, athletes are placed under extreme physiological pressure and oftentimes enter into large energy deficits. This may result in underperformance, evident through a variety of physical and biochemical markers such as a reduced time to exhaustion and muscle glycogen sparing, respectively [5,6]. Over the previous three decades, the popularity of ultra-endurance activity, specifically ultramarathon running, has continued to gain momentum [7,8]. Successfully completing such events requires long-term preparation, sufficient exercise management, supportive dietary habits, nutritional intakes to accommodate optimal adaptations, and psychological strength [9]. Ultra-events can be classified as either semi-supported, whereby event organizers provide ad libitum food and fluids, or self-sufficient, whereby participants are required to transport all necessities with them throughout the race [10]. The latter usually involves challenging terrain, extreme environmental conditions, and rough sleeping situations [11,12,13].

Potentially the most common cause of underperformance during endurance events, exercise-related gastrointestinal symptoms (GIS) are estimated to affect between 30 and 90% of distance athletes [14]. GIS such as the urge to regurgitate, regurgitation, bloating, belching, cramping, and increased flatulence are commonly reported by participants in strenuous exercise, whereby prolonged activity influences an athlete’s physiology, resulting in significant changes evident through biochemical marker measuring [15,16]. In ultramarathon events specifically, GIS is often cited as a leading reason for failing to complete the course [17]. Physiological, mechanical, psychological, and nutritional pathways may increase GIS [18]. Oftentimes, this is due to a decrease in gastric motility and emptying, an increase in bacterial translocation, and altered intestinal permeability, resulting in exercise-induced nutrient malabsorption [19]. The effects of this are then exacerbated by extreme environmental conditions, such as the combination of high heat and humidity, which, through limiting the evaporation of sweat, impairs thermoregulation, or high altitude, which may damage the intestinal barrier [20,21,22].

While the majority of exercise-induced GIS are mild and pose non-existent to minimal long-term health risks, serious gastrointestinal health conditions may still arise [23]. These include, but are not limited to, hemorrhagic gastritis, whereby the lining of the stomach (mucosa) becomes inflamed, and ischemic bowel disease, occurring when the blood flow through major arteries in your intestines slows or stops to exist, as a few of the more challenging health complications as a result of such highly taxing forms of exercise [24]. Classified using the Rome criteria, Irritable Bowel Syndrome (IBS) is a debilitating and chronic functional gastrointestinal disorder characterized by abdominal pain in addition to an alteration in bowel habits, whereby there is an absence of detectable physical or biochemical abnormalities [25,26]. IBS can be categorized as either diarrhea-predominant (IBS-D), constipation-predominant (IBS-C), existing with alternating or mixed stool patterns (IBS-S), or unclassified IBS [27]. Previously considered a functional gastrointestinal disorder, but more recently re-defined as a disorder of the gut-brain interaction, IBS is a symptom-based diagnosis established through a patient recording of their experienced symptoms [28]. Unlike a syndrome, a symptom is a perceived experience differing from those recognized as normal, while a syndrome is a consistent group of symptoms [29]. Though the etiology of IBS is poorly understood, its pathogenesis has been associated with a plethora of conditions. These include, but are not limited to, altered gastrointestinal motility, the overgrowth of bacteria in the gastrointestinal tract, visceral hypersensitivity, and intestinal inflammation [30]. There are believed to be many mechanisms active at a molecular level that are linked to IBS. For instance, the release of serotonin in plasma has been observed to be increased in individuals with IBS-D but reduced in those with IBS-C [31]. Additionally, psychological factors have been observed to play a significant role in IBS, with issues such as depression and anxiety common in those with an IBS diagnosis [32,33]. However, the exact causes have yet to be adequately identified [30]. Though exercise is advocated for general health and wellbeing, research shows that even with exercise of moderate intensity (3.0–5.9 metabolic equivalents), the gastrointestinal tract may be compromised and subsequent GIS experienced [34]. With ultra-endurance activity considered strenuous at ≥6 metabolic equivalents, these athletes are at an increased risk of suffering from exercise-induced GIS [15]. Regarding the prevalence of diagnosed IBS amongst endurance and ultra-endurance athletes, Killian and Lee [35] noted that while the total prevalence of medical diagnoses in their cohort of *n* = 430 participants was 9.8%, over 56% of athletes experienced at least one GIS. They concluded that, while not medically diagnosed, these athletes could potentially benefit from IBS GIS mitigating strategies.

Nutritional interventions continue to be a major tool used by athletes to enhance psychological, skeletal, muscular, and cardiovascular system performances [36,37].The high energetic demands that accompany long-duration activity make these interventions of particular importance to ultra-endurance athletes [38]. Endurance athletes have been found to employ a variety of nutrition strategies to manage exercise-induced GIS [39]. However, the data on such practices within an ultra-endurance population remains to be compiled, making it a challenge for sport nutrition professionals to develop supportive recommendations. Regarding pre-race nutrition, endurance athletes are typically advised to follow the general sports nutrition guidelines of avoiding high-fat, high-protein, and high-fiber foods [40]. In addition to this, they are advised to establish their own individual food intolerances and employ strategies to avoid them, resulting in minimal GI upset [41,42]. Issues arise, however, when athletes eliminate specific foods or food groups with the belief that these foods/groups are the primary cause of their exercise-induced GIS. For example, following a gluten-free diet has anecdotally gained popularity within this population for minimizing GIS. Adopting this diet has been observed in up to 43% of non-celiac athletes as a method for minimizing GIS. Non-celiac athletes who adhere to such a diet are at increased risk for numerous negative issues, such as failing to meet optimal nutrient intakes, an increase in food costs, and a potential alteration in beneficial gut bacterial populations [43,44]. With regards to failure to ingest adequate nutrients and its accompanying energy deficit, there is a subsequent increased risk of suboptimal performance, such as a reduction in time to exhaustion and inadequate exercise recovery [45]. However, the majority of these athletes rationale for adopting this gluten-free dietary pattern is not based on medical advice but driven by the perception that a diet free from gluten is superior to one containing gluten for attenuating GIS [46]. In a similar fashion, endurance runners have been shown to regularly avoid foods like meats, milk products, seafood, poultry, and foods that are high in fiber for similar reasons [40]. Comparably, evidence exists supporting the potential benefit of some dietary strategies, such as diets low in fermentable oligosaccharides, disaccharides, monosaccharides, and polyols, for reducing exercise-induced GIS [47]. While investigating the dietary patterns of both runners diagnosed with IBS and undiagnosed with IBS but suffering reflux frequently, [48] found that while many athletes avoid triggering foods in pre-race meals, they also consume foods that are potentially aggravating their GIS and would likely benefit from further nutritional advice for those suffering with IBS or IBS-related symptoms.

Taking into consideration the multifaceted demands and stressors of ultra-endurance activity, including the accompanying exercise-induced GIS, the necessity for individualized pre-, during, and post-race nutritional strategies is clear. Recognizing that the vast majority of nutritional recommendations for ultra-endurance athletes are derived from research employing shorter distances, it is crucial that more attention be devoted specifically to those undertaking exercise for a minimum duration of 4–6 h.

With this in mind, the aim of this systematic review is to explore the existing research that employs nutritional strategies before, during, or after exercise in an attempt to alleviate exercise-induced GIS in ultra-endurance athletes. For the purpose of this research, studies conducted on individuals who were considered to be “ultra-endurance athletes” by the authors were the defining criteria for inclusion in this review.

## 2. Methods

### Study Design

The PICOS (population, intervention, comparators, outcomes, study design) model for this research was employed for the definition of the inclusion criteria described in Table 1 [49]. Scientific literature was systematically searched to investigate the current nutritional recommendations and common practices of trained ultra-endurance athletes to alleviate GIS. An overview of the search topics and criteria can be found in Appendix A.

A systematic literature search was conducted through the Medline (PubMed), CINAHL, Web of Science (WoS), and Sports Discus databases in November of 2021 and again in January of 2023 to support the development of this review (Table 2). Key words were defined in accordance with the following terms: ultra-endurance; exercise; gastrointestinal; nutrition; intervention; symptoms. These, along with similar words (see Appendix A) were searched using Boolean operators. Any original article reporting on interventions to alleviate GIS resulting from ultra-endurance exercise and occurring in a population defined by the researchers as “ultra-endurance athletes” was considered for analysis. Included studies had to report on GIS (as defined and measured by the authors as symptoms, severity, and/or duration) as an outcome measure. Only articles reporting on healthy humans (without a diagnosed medical condition, with the exception of IBS) aged ≥18 years who partook in ultra-endurance exercise were included. 

Interventions and Comparators: Any original article reporting on nutritional interventions to alter participants’ GIS as a result of ultra-endurance exercise was considered for analysis. There were no filters applied to intervention length, blinding procedure, or comparators. Both “No comparators” and “placebos as comparators” were acceptable.

Inclusion and Exclusion Criteria:

The inclusion criteria applied during the selection of suitable studies included the following:

i. Studies conducted on individuals who were considered to be “ultra-endurance athletes” by the authors;

ii. Studies that include a nutritional intervention with the aim of reducing ultra-endurance exercise-induced GIS;

iii. Studies that include GIS as an outcome of interest;

iv. Studies with the full text freely available in the English language.

The exclusion criteria applied to the populations and protocols of this review included the following:

Studies conducted on subjects with a diagnosed medical condition, excluding IBS;

a. On subjects who were currently, or have been within the previous year, injured (to the extent of halting training), pregnant or given birth, or undergone surgery;

b. On subjects consuming, or that have consumed within the previous three months, medication that may alter GIS;

c. With non-nutritional interventions;

d. With no measurement of GIS incidence, severity, or duration;

e. Which were not freely available in the English language.

There were no filters applied to participants’ race, gender, or ethnicity in order to increase the analytical power of the analysis.

Study Selection and Data Extraction: Articles from this search underwent multiple revisions prior to being accepted for inclusion in this piece of work and were reviewed by independent members of the research team, LR and ED. Searches were downloaded and saved to Endnote 20.2.1. (Endnote, Philadelphia, PA, USA). A total of 1528 papers were identified for possible inclusion and further investigated for suitability. Firstly, titles were examined for relevance to the research topic. Following this, abstracts were evaluated for relevance. Finally, the methodology of each study was assessed for suitability in this review. The search for relevant published articles was carried out by an individual author (T.R.) and reviewed independently by two authors (L.R. and E.D.), with any disagreements resolved through discussion.

Data Analysis: Once the inclusion and exclusion criteria had been applied to each individual study, information on the study source (authors and year of publication), study type, mode of intervention delivery, dosage and timing, population sample size, characteristics of participants (demographics, ultra-endurance exercise experience), and the final outcomes were extracted by one individual author, T.R., to a Microsoft Excel spreadsheet (Microsoft Excel, version 2207). Due to the heterogeneity between the included study protocols, no statistical analysis was conducted.

Outcome Measures: Studies that met the previously discussed criteria as well as contained an outcome measure related to GIS (incidence, duration, and/or severity) were accepted for inclusion. Where relevant, the statistical significance of the findings is given, recognized as *p* < 0.05.

Quality Assessment of the Experiments: The identified studies were assessed for methodological quality and bias risk in accordance with the Academy of Nutrition and Dietetics Quality Criteria Checklist for Primary Research by a single author, T.R. [50]. This tool has been commended for its high inter-observer agreement, time efficiency, and ease of use [51]. Disagreements were resolved by discussion, if necessary, with the paper quality graded as positive (+), negative (−), or neutral (/), as illustrated in Table 3. Only two of the seven studies mentioned the use of a power calculation for their research [52,53].

## 3. Results

### 3.1. Flow Diagram of Studies Retrieved for the Review

Unrelated articles were excluded on the basis of title, then abstract, by one author (TR), with results independently reviewed by two authors (LR and ED) and conflict settled through group discussion. Articles deemed eligible for full-text review were retrieved and screened by one author (TR), with results independently reviewed by two authors (LR and ED) and conflict settled through group discussion (Figure 1).

### 3.2. Study Selection and Characteristics

The majority of the studies included in this systematic review employed a crossover intervention design, with one parallel-group intervention, one counter-balanced intervention, and a single field-based case study with the characteristics of each tabulated (Table 3).

The most frequently utilized rating scale for GIS ranged from 0 to 10. The smallest scale ranged from 0 to 3, while the largest ranged from 0 to 100. Visual Analogue Scales (VAS), Likert-type scales, and BORG or modified BORG scales were commonly employed to record GI outcomes. For these measurements, the larger the number, the more severe the athlete considered their GIS. One study did not use a scale and instead recorded GIS as “number of healthy days” and “number of GI episodes”. The studies in this review included a total of *n* = 105 participants, of which *n* = 50 participants were male and *n* = 55 participants were female.

## 4. Synthesized Findings

### 4.1. Dietary Intervention: Low FODMAP

Gaskell and Costa [54] recognized the potential of a low-in fermentable oligo-, di-, and monosaccharides and polyols (FODMAPs) for GIS management, as well as the increased need for such a tool amongst ultra-endurance athletes diagnosed with Irritable Bowel Syndrome (IBS). In their case study of an individual female recreational ultra-endurance runner diagnosed with IBS and partaking in a multistage ultramarathon, they investigated the impact of the low-FODMAP diet on exercise-associated GIS [52]. A nutrition plan was developed by an accredited sports dietician in accordance with the participants intolerances and specific requirements. The intervention was applied six days prior to the 186.7 km mountainous multistage race and maintained for the duration of the event (six days). A 100 mm visual analog scale (VAS) was utilized in order to quantify GIS incidence and severity, while nutritional intake was analyzed with FoodWorks Professional Edition, Version 6.0.2539 (Xyris Software, Brisbane, Australia) and compared with current recommendations. The runner successfully followed the low-FODMAP dietary protocol, with a reduction in GIS severity from severe during training sessions (VAS: 60–100 mm) to modest during the event (VAS: 30 mm) and a vast reduction in the variety of GIS experienced. However, the runner experienced daily moderate to severe nausea, which the authors suggested was potentially due to her failure to meet carbohydrate intake and, thus, energy intake recommendations. While the runner was satisfied with her IBS symptom management prior to and during the race, she was dissatisfied with the nausea she experienced and its subsequent effect on her nutritional intake and potential performance negation. This case study supports the implementation of a low-FODMAP diet for IBS symptom control in diagnosed ultra-endurance athletes. This is a single-case study with inherent generalizability limitations, but it does provide valuable insights, especially for the niche population of ultra-endurance runners.

Further investigating the effects of FODMAPS on exercise-induced GIS, Gaskell et al. [52] employed a randomized double-blind crossover trial to determine the effects of short-term, 24 h low- and high-FODMAP diets on GIS resulting from exertional heat stress. They recruited *n* = 18 endurance and ultra-endurance runners, of which *n* = 10 were male and *n* = 8 were female. One week prior to the experimental trial, all participants completed a continuous incremental exercise test for volitional exhaustion on a motorized treadmill in order to estimate their maximum oxygen uptake. This was then used to determine the treadmill speed for the experimental trials, which was 60% of maximum oxygen uptake, while the gradient was set to 1%. Employing a double-blind randomized cross-over design, participants were provided with either a diet high (HFOD) or low (LFOD) in FODMAPs for the following 24 h period prior to each experimental trial. Blinding was successfully achieved by supplying foods that were sealed and vacuum packaged, making them similar in physical appearance. Though the HFOD and LFOD meals were the same, through manipulating the ingredients, their FODMAP content differed. The food for the participants was prepared by an external researcher, a dietician, further strengthening the blinding process. On the day of each trial, participants arrived in the morning after consuming their assigned breakfast 1 h prior. At this point, they had completed their first 10-point modified VAS GIS assessment tool. The rating scale was as follows: 1–4 is indicative of mild GIS, which was not substantial enough to interfere with exercising; 5–9 is indicative of severe GIS, which was substantial enough to interfere with exercising; and 10 is indicative of extreme GIS resulting in exercise cessation. Participants then completed a steady-state run on a motorized treadmill at their previously determined speed in an environmental chamber, which lasted for a duration of 2 h. During this period, GIS measurements were collected every 15 min, with the final measurement taken immediately after exercising. Participants then consumed their assigned high- or low-FODMAP recovery beverage and remained seated for a 4 h recovery period in which GIS continued to be measured at 15 min intervals. On the HFOD, the severity of total- and lower-GIS was higher pre-exercise in comparison with the LFOD (*p* = 0.017 and *p* = 0.074, respectively). Similarly, GIS was higher during exercise following the HFOD diet than the LFOD diet. Likewise, during the entire HFOD trial, gut discomfort severity (*p* = 0.056), total (*p* = 0.014), upper (*p* = 0.019), and lower (*p* = 0.006) were higher than during the LFOD trial. The authors noted that there was no significant difference for any of these GIS-related measurements between trials when completing a sub-group analysis for the male and female participants. These findings further support the notion that modifying dietary FODMAP content prior to and during exercise may positively influence the GIS of healthy endurance and ultra-endurance athletes prone to exercise-associated GIS.

### 4.2. Supplementation: Lactose-Free Pre-Exercise Meal

Over 65% of the global population is estimated to suffer from some form of lactose malabsorption or intolerance [55]. For individuals who are diagnosed with IBS or suffer from frequent IBS-like symptoms, the exclusion of lactose-rich foods and beverages is well-established as a successful strategy to alleviate GIS [56]. Haakonssen et al. [53] recruited *n* = 25 healthy competitive female cyclists for their randomized, counter-balanced, crossover intervention investigating the effect of a dairy-based pre-exercise meal on gut comfort. Participants comprised of *n* = 25 cyclists registered with the Australian National Road Series teams, an international professional cyclist, an ultra-endurance mountain biker, and five well-trained national club-level cyclists, all with 18 months of racing experience. For the 24 hr period before each trial, participants’ diets were standardized to provide 5 g per kg body mass (BM) carbohydrate, 1.5 g per kg BM protein, and 1.5 g per kg BM fat. Meals were supplied pre-packaged, with pre-trial meals provided in the laboratory. Pretrial meals were scaled to provide carbohydrates at 2 g per kg BM. The dairy-based meal consisted of rolled oats cooked with calcium-fortified milk, yogurt, and additional milk, while the control meal consisted of oats cooked in water, canned fruit, and some nuts. Although the two meals looked similar, complete blinding of participants was not possible due to the difficulty of concealing the variation in dairy content. The exercise protocol was conducted in a laboratory and consisted of an 80 min ergometer cycle, followed by a 10 min time trial. For all groups, a carbohydrate gel was consumed at the 30- and 60-minute mark, with hydration at libitum for the first trial and then replicated for the second trial 48 h later. GIS were recorded at five separate timepoints (before the pre-trial meal, 30- and 60-minutes after the meal, immediately pre- and post-exercise) by answering how their stomach felt at that moment according to a 5-point Likert-type scale from 1 (very comfortable) to 5 (very uncomfortable). Participants were informed to answer questions in relation to GIS, such as nausea, bloating, and pain. Although results suggest that the dairy-based pre-exercise meal was more palatable, the authors observed no statistically significant associations between pre-trial meal type, whereby the lactose content differed, and gut discomfort (*p* = 0.15), or between pre-trial meal type and gut discomfort scores at the 30- or 60-minute mark (*p* = 0.31; *p* = 0.17). The lack of full blinding in this study due to meal differences has the potential to introduce bias. As the results are non-significant, it is possible that lactose might not play a significant role in GIS for the population studied.

Russo et al. [57] examined the impact of compositionally different, flavored dairy milk-based recovery beverages on GIS in recreationally and competitively trained endurance and ultra-endurance runners. The authors recruited *n* = 9 participants, of which *n* = 7 were male and *n* = 2 were female, for their randomized cross-over study consisting of two experimental laboratory-based trials separated by a minimum of 5 days. For the 24 h prior to and throughout the experimental trials, athletes consumed a diet low in FODMAPs (<2 g FODMAPs per meal), with compliance monitored through the use of a food and exercise log. In a randomized order, participants were assigned to two experimental trials, which were separated by a minimum 5-day washout period. For one trial, participants attended the laboratory after consuming a standardized low-FODMAP breakfast. Prior to beginning exercising, they completed an exercise-specific modified VAS. Following this, they completed 2 h of a high-intensity interval exercise session at 23.4 ℃ and 42% relative humidity. GIS measurements were recorded during the final 30 s of exercising. Thirty minutes after exercise cessation, the recovery period began. During this time, participants rested in a supine position while GIS were recorded at 30 min intervals for the following 4 h. Two hours into the recovery period, athletes were supplied with a standardized low-FODMAP meal and instructed to consume as much as they were able to. In the evening after leaving the laboratory, participants consumed another standardized low-FODMAP meal. The following morning, athletes attended an exercise performance assessment in the laboratory. Similar to the previous day, they received a standardized low-FODMAP breakfast. GIS were measured before and directly after the performance test, which comprised a 20 min running exercise to measure maximum oxygen uptake and oxidation rates at four submaximal intensities, followed by a 1 h performance running test on a motorized treadmill. In a randomized, repeated-measures design, participants received a recovery beverage of either a commercially available chocolate-flavored dairy milk (CM, 1.2 g/kg BM carbohydrate and 0.4 g/kg BM protein) or a commercially available chocolate-flavored dairy milk-based supplement beverage (MBSB, 2.2 g/kg BM carbohydrate and 0.8 g/kg BM protein). Beverages were matched to volume and supplied in three equal boluses at 10 min intervals beginning 30 min into the recovery period, served in opaque bottles. Participants were instructed to drink as much as was tolerable. The authors observed a main effect of time for lower GIS at 3.5 h (*p* > 0.01) and 4 h (*p* < 0.05) into the post-exercise recovery period. Additionally, there was a trend towards increased total gut discomfort on the MBSB trial (*p* = 0.053), but there were no significant main effects observed for nausea, upper- or total-GIS. The findings of this paper suggest that MBSB consumption results in carbohydrate malabsorption and, subsequently, an increased trend towards experiencing greater gut discomfort. This contradicts previous findings and may be due to a variety of factors. Most notably, participants in this study followed a low-FODMAP diet prior to and during the trial. Their FODMAP intakes, in addition to the FODMAPs in their assigned interventions, may collectively induce GIS. In addition to this, the GIS rating scales and protocols varied [56]. However, it is important to recognize the small sample size and unequal male-to-female ratio of this research, which may influence the generalizability of its findings.

### 4.3. Supplementation: Medium-Chain Triacylglycerol and Carbohydrates

Goedecke et al. [58] investigated, by means of a randomized, placebo-controlled crossover study, the effect of medium-chain triacylglycerols (MCTs) in combination with carbohydrates on substrate metabolism and ultra-endurance exercise performance. A total of *n* = 8 endurance-trained cyclists were recruited to partake in two separate 270 m cycles at 50% of peak power output interspersed with four sprints at 60 min intervals, followed by a 200 kJ time trial in a laboratory setting. Participants were randomly assigned to receive either 75 g of carbohydrates or 32 g of MCT pre-trial, then a 10% carbohydrate solution or 4.3% MCT and 10% carbohydrate solution, respectively, during exercise. Trials were separated by a minimum of seven days. The majority of subjects recognized a difference in mouthfeel but were not made aware of the composition of either drink. Participants previous experiences with MCT were not recorded. GIS incidence and severity were recorded using a scale from 0 to 3, with 0 corresponding to no symptoms and 3 to symptoms severe enough to impact performance. Half of the participants reported suffering from GIS, the majority of whom considered their symptoms “severe”, during or after the MCT trial. In contrast, there was a complete lack of GIS experienced during the carbohydrate trial. The main findings of this study suggest that MCTs ingested with carbohydrates have a detrimental effect on ultra-endurance sprint performance. The authors noted that this effect could potentially be due to the GIS being more prevalent with MCT ingestion. However, as participants previous experiences and exposures were not recorded, the GIS experienced may be due to participants being unaccustomed to MCTs.

### 4.4. Adaption: Gut Training for Carbohydrate Tolerance

Investigating the adaptability of the gastrointestinal system to improve carbohydrate tolerability, Costa et al. [59] recruited *n* = 25 recreationally competitive endurance and ultra-endurance runners not diagnosed with IBS for their parallel-group trial involving two different gut-training protocols. Participants consumed a standardized low-fiber and low-FODMAP breakfast with water, then reported to the testing center. Participants reported GIS using a previously developed 10-point Likert-type scale (0 indicating no symptoms, 10 indicating extreme symptoms) within 30 min before undertaking their assigned trial [60]. Measurements were categorized as gut discomfort, upper-, lower-, or other-GIS, with participants in trial 1 having the highest GIS at baseline. Participants completed a 120 min steady-state run, followed by a 60 min distance test with water ad libitum. During exercise, participants completed gut training protocol 1, while they ingested 30 g of carbohydrate at a 2:1 glucose/fructose ratio gel-disc supplement at 0 min and every 20 min thereafter. GIS was recorded in 10 min intervals. Following this, they were randomly assigned to one of three protocols: 1. an amount of 30 g of carbohydrate gel-disc supplementation during running for 10 days over 2 weeks; 2. A low-carbohydrate placebo-matched formulated gel-disc (Sugarless Liquid, Bathox Australia, Chipping Norton, NSW, Australia); and 3. An amount of 30 g of carbohydrates from a food portion (AmazeBalls Original, Runners Kitchen, Melbourne, Victoria, Australia). After completing their trial along with 2 days of rest, participants returned to the laboratory and completed the same gut trial as previously described. Normal diets were consumed during protocols, with intake and exercise monitored for three days before the initial trial and throughout protocols. Although the total accumulation of points for GIS and their severity did not significantly differ between groups in trials 1 and 2, significant improvements for all GIS-related outcomes were observed with the carbohydrate supplement and food compared with the placebo. These findings support the notion that gut training has the potential to improve carbohydrate tolerability, regardless of supplement form. However, the results are truncating and do not provide definitive, clear outcomes or implications.

In a similar piece of work, Miall et al. [61] recruited *n* = 18 competitive endurance and ultra-endurance runners for their randomized controlled trial of a gut-training protocol. Within 30 min of beginning their initial trial, participants recorded GIS using a 10-point Likert-type rating scale, where 0 indicated no symptoms, 5 indicated severe symptoms, and 10 indicated extreme symptoms. These GIS were categorized as overall gut discomfort, upper-, lower-, and other-GIS. Following this, participants performed the initial gut challenge involving 120 min of running at 60% VO_2_ max while consuming a 2:1 glucose/fructose 30 g carbohydrate gel-disk at 0 min and then every 20 min. GIS were measured every 10 min. This was followed by a 60 min running effort where they were instructed to run “as far as possible” within the timeframe. Upon completion, participants were randomly assigned to a blinded 90 g/hr. carbohydrate or 0 g/hr. placebo gut-training protocol. They consumed their assigned supplement during 1 h runs daily for the next fortnight. They then repeated the protocol for gut challenge 1. The authors found that, in comparison with placebo, 2 weeks of carbohydrate supplementation reduced gut discomfort (*p* = 0.008), total (*p* = 0.009), upper (*p* = 0.15), and lower (*p* = 0.008) GIS. This information further strengthens the findings of Costa et al. [59] suggesting that carbohydrate adaptation is a potential method for improving its tolerability during endurance exercise. 

### 4.5. Assessment and Risk of Bias

Methodological quality assessment was performed in accordance with the Academy of Nutrition and Dietetics Quality Criteria Checklist (ADA, 2008) for Primary Research for all the studies that met the defined inclusion criteria, and the results of the quality analysis did not alter the decision of whether to include an article in this review as all received a positive score.

## 5. Discussion

### 5.1. Summary of Main Findings

#### Low FODMAP

Multiple of the seven studies included in this review investigated the impact of FODMAPs on GIS. These quickly fermented, short-chain carbohydrates are common triggers of GI upset due to their ability to increase both intraluminal gas and osmotic pressure, particularly in viscerally hypersensitive individuals [62]. With up to 23% of athletes meeting the criteria for IBS diagnosis and many more experiencing IBS-like symptoms, studies recruiting such populations are highly warranted in order to support professionals in their development of appropriate dietary recommendations [35]. Restricting the intake of FODMAPs has proven a successful method of alleviating GIS in approximately 70% of individuals with a diagnosis. Through the osmotic pressure they create, water is forced into the gastrointestinal tract [63]. Additionally, they are easily fermentable in the colon and subsequently increase gas production. Such activity results in undesirable GIS, such as bloating and discomfort. Such issues are not only experienced by those with a clinical diagnosis. For example, fructose is absorbed in the small intestine but may be inefficiently absorbed when excess glucose is present or, as is more so the case for athletes, as a result of a reduction in intestinal transit time caused by exercise [64]. Although there is emerging evidence in support of the low FODMAP diet for GIS management in those diagnosed with IBS, the vast majority of research has focused on non-athletic populations [65,66]. In their case study of an individual female recreational ultrarunner diagnosed with IBS and partaking in a multistage ultramarathon, Gaskell and Costa [52] investigated the impact of a diet low in FODMAPS on exercise-associated GIS [52]. A nutrition plan was developed by an accredited sports dietician in accordance with the participants intolerances and specific requirements. The intervention began six days before and was maintained for the duration of the race. The low-FODMAP diet reduced symptoms from severe during training sessions to modest during the event and substantially lowered GIS incidence. Although the runner experienced daily nausea, this was attributed to an insufficient energy intake. These findings support the emerging evidence in support of the low-FODMAP diet for GIS management. In a similar case study, Lis et al. [67] implemented a short-term low-FODMAP dietary intervention in a recreationally competitive multisport male who had no GI-related diagnoses but was prone to suffering from GIS. The effect of a six-day low-FODMAP diet on GIS was examined in comparison with six days of the athlete’s habitual dietary pattern. Symptoms were recorded immediately post-exercise and at the end of each day on a scale ranging from 0 to 9. Comparable with the findings of Gaskell and Costa, the low-FODMAP dietary protocol effectively combatted the athlete’s GIS, with GIS ranging from 0 to 4 during his normal habitual diet, reduced to 0 (no symptoms at all) for all measurements during the low-FODMAP diet phase. In 2018, Lis et al. [68] recruited *n* = 11 recreationally competitive runners (running over 25 km per week) with a history of nonclinical exercise-associated GIS for their single-blinded cross-over study. Participants were allocated a high- or low-FODMAP diet for six days, followed by a single-day washout period. During each phase, the participants completed 5 × 1000 m runs and a 7 km threshold run. GIS were recorded during exercise and daily, with results showing a significantly lower level of daily flatulence (*p* < 0.001), urge to defecate (*p* = 0.004), loose stools (*p* = 0.03), and diarrhea (*p* = 0.004) during the low-FODMAP dietary phase but no significant difference in during-exercise GIS. Also employing a crossover design, Wifflin et al. [47] investigated the effect of seven days of a low- or high-FODMAP diet on exercise-related GIS during training among *n* = 16 recreational runners. Recording symptoms with an IBS-Severity Scoring System questionnaire, the authors observed a significant reduction in participant GIS during the low-FODMAP diet (*p* = 0.004).

Also included in this review, [69] employed a randomized double-blind crossover trial to determine the effects of short-term, 24 h low- and high-FODMAP diets on GIS in response to exertional heat stress. *n* = 18 endurance and ultra-endurance runners participated, assigned to one of the dietary strategies, before completing two hours of running at 35 degrees Celsius at 60% VO_2_ max. GIS were measured pre-exercise, at 15 min intervals during exercise, immediately after exercise, and throughout the recovery period. Similar to previous research in this area, the authors noted greater GIS severity experienced by athletes during the high-FODMAP trial. Although investigating carbohydrate intake and GIS in endurance exercise, King et al. [70] did not control their research for FODMAP intake, noting that in not doing so they allowed for the “real-world” dietary habits of participants to continue. While laboratory-based research is warranted to clearly identify the potential of FODMAP manipulation on endurance-exercise-induced GIS, current research suggests that lowering one’s FODMAP intake is a promising approach. Additionally, further investigating the “real-world” diets of ultra-endurance athletes will enable nutrition professionals to develop practical dietary advice.

### 5.2. Lactose-Rich Pre-Exercise Meal

Avoidance of milk products is among the most popular dietary restrictions for endurance athletes to attenuate GIS [40]. In a self-assessment of nutritional habits, 62.5% of *n* = 40 female middle- and long-distance runners described their diet as “outstanding”. These runners had a significantly lower intake of dairy beverages in comparison with their self-proclaimed “average” diet counterparts, as well as an overall significantly lower intake of lactose, a known FODMAP (*p* = 0.0119). For individuals who are diagnosed with IBS or suffer from frequent IBS-like symptoms, the exclusion of lactose-rich foods and beverages is well-established as a successful strategy to alleviate GIS [56]. Issues arise when individuals without intolerances take it upon themselves to remove entire food groups, risking suboptimal nutrient intakes and increased food costs [71]. Milk is an affordable and easily accessible source of calcium, an essential mineral associated with interleukin-8 and the inflammatory response. Sweating induced by heat stress, especially during exercise, promotes hypohydration and a loss of electrolytes, including calcium [72]. The resulting osmolality impairment promotes an increase in reactive oxygen species [73,74]. Consequentially, there is a rise in systemic inflammation within the body [75]. Haakonssen et al. [56] observed no statistically significant associations between pre-trial meals and gut comfort (*p* = 0.15) during their investigation of such a dietary alteration on exercise-induced GIS under laboratory conditions. This study was the first of its kind to investigate the effects of a dairy-based pre-exercise meal on gut comfort and GIS of female cyclists (*n* = 25 cyclists registered with the Australian National Road Series teams, an international professional cyclist, an ultra-endurance mountain biker, and five well-trained national club-level cyclists, all with 18 months of racing experience). The authors noted that they did not account for the history of GIS in their participants, which has been proven to strongly correlate with exercise-induced GIS [76]. The potential of dairy-rich pre-exercise meals to enhance or impair gut comfort during prolonged exercise remains unclear. While research in this area is sparse, this study’s findings support the inclusion of a substantial amount of dairy in the meal prior to strenuous cycling without a negative impact on gut comfort.

Both recreational and elite endurance athletes frequently enlist the help of sports-focused foods and supplements to support exercise recovery [77]. Many beverages specifically formulated to promote optimal recovery are dairy-based and align closely with the guidelines and recommendations for post-exercise nutrition (1.0–1.2 g·kg^−1^ BM of carbohydrates and 0.2–0.4 g·kg^−1^ BM of protein) [78]. In recognition of these guidelines, these beverages often provide high energy and nutrient content, specifically carbohydrates and protein. Included in this review, [57] investigated the effect of compositionally different, flavored dairy milk-based recovery beverages on GIS in both recreationally and competitively trained endurance and ultra-endurance runners. For the 24 h prior to and throughout the experimental trials, athletes consumed a diet low in FODMAPs (<2 g FODMAPs per meal). In a randomized order, participants were assigned to two experimental trials whereby they received a post-exercise recovery beverage of either a commercially available chocolate-flavored whole food dairy milk [54] or a commercially available chocolate-flavored reconstituted dairy milk-based supplement beverage (MBSB). The authors noted a trend towards increased total gut discomfort on the MBSB trial (*p* = 0.053), but there were no significant main effects observed for nausea, upper- or total-GIS between trials. The results of this paper suggest that MBSB consumption during the recovery phase, with its increased nutrient composition, results in carbohydrate malabsorption and, subsequently, an increased trend towards experiencing greater gut discomfort. The authors concluded that while this research provides evidence of how nutrition and exercise-induced factors may contribute to the malabsorption of dairy-based recovery beverages, further research is needed to define the underlying mechanisms at play in order to support the development of counteracting recommendations.

### 5.3. Medium-Chain Triacylglycerol and Carbohydrates

MCTs are shorter-chain fatty acids that have been heavily investigated as an ergogenic aid for endurance exercise [79]. The basis for this is its ability to alter substrate metabolism. While small amounts (25–30 g) fail to have any obvious effects, ingestion of large boluses (45 g) has resulted in increased fat oxidation [80,81]. MCTs are broken down to glycerol and medium-chain fatty acids and do not reduce gastric absorption or emptying. Instead, the resulting fatty acids are metabolized at a similar rate to that of glucose, suggesting fat may be a potential fuel source alternative during exercise [82]. A high intake of MCT may be detrimental to performance due to its impact on GIS prevalence [83,84]. Ref. [58] investigated MCT ingestion in combination with carbohydrates on GIS amongst ultra-endurance runners during a laboratory-based exercise protocol. A total of 50% of participants reported suffering from GIS during the MCT trial, the majority of which were considered “severe”, while no GIS was experienced during the carbohydrate trial. Consistent with other research, nausea, cramps, bloating, vomiting, and diarrhea were the most commonly experienced symptoms [79]. However, it is important to recognize the setting of this research. As this work was conducted under laboratory conditions, its applicability in the real world has yet to be determined. Current recommendations acknowledge the continuous debate regarding high-fat diets for endurance performance. A strategy gaining momentum is that of “train low, compete high”, involving dietary periodization of training in a high-fat, low-carbohydrate state, then re-introducing carbohydrates into the diet before race day. While performance benefits have been observed with higher fat intakes, there remains too much controversy to establish definitive guidelines [85].

### 5.4. Repetitive Gut Challenges

A highly adaptable system, the GI tract may potentially be trained to improve certain tolerabilities during exercise, such as carbohydrate tolerability [86,87]. With severe GI symptoms reported by up to 85% and 73% of multi-stage and single-stage ultra-distance runners, respectively, supporting such adaptation would be extremely beneficial for this population [60]. Costa et al. [59] recruited participants who completed an initial gut trial consisting of a run on a motorized treadmill under laboratory conditions. Following this, they were assigned to one of three gut-training protocols. For then until returning to the laboratories for their second experimental trial, participants were informed to continue consuming their usual diet along with their assigned formulated gel-disc. For monitoring compliance, participants were asked to complete a gut-training adherence and completion log throughout their gut-training period. They found that GI distress and GIS did not significantly differ between groups of recreational and competitive ultra-endurance runners who did and did not undergo 2-weeks of repetitive carbohydrate gut challenges. In a similar piece of work by Miall et al. [61], *n* = 18 competitive endurance and ultra-endurance runners took part in a randomized controlled trial of a gut-training protocol. In comparison with placebo, two weeks of carbohydrate supplementation reduced gut discomfort (*p* = 0.008), total (*p* = 0.009), upper (*p* = 0.15), and lower (*p* = 0.008) GIS. This information further strengthens the findings of Costa et al. [59] suggesting that carbohydrate adaptation is a potential method for improving its tolerability during endurance exercise. In a 2022 study, King et al. [88] investigated the effects of a short-term high-carbohydrate diet and gut training on GIS in trained endurance athletes. Their multi-pronged strategy consisted of 2 weeks of high carbohydrate ingestion and gut training followed by acute strategies to promote endogenous and exogenous carbohydrate availability, while their comparator consisted of strategies reflecting lower ranges of current guidelines [89]. This strategy was associated with increased GIS in the second half of a 26 km walk (*p* = 0.970), particularly upper GIS (*p* > 0.01), compared with CON. Participants also completed a 35 km run. Post-intervention, lower- and peak-GIS showed significant increases (*p* = 0.016 and *p* = 0.002, respectively). However, no differences were observed between dietary groups for any variables (*p* > 0.05). The authors concluded that, with practice, increased carbohydrate intakes can be tolerated by athletes and may enhance athletic performance without major attenuations of GI comfort.

### 5.5. Limitations

Given the nature of the articles in this review, the generalizability of these findings is limited. The research included in this review examined the effect of various nutritional interventions on GIS experienced by ultra-endurance athletes, though the quantity of acceptable papers for inclusion was low. Only two of the seven studies in this review discussed employing a power calculation to support their chosen sample size. Studies that recruited endurance athletes but did not state if some were ultra-endurance athletes were excluded. Additionally, articles not freely available in the English language were excluded, which may leave some research overlooked. With this in mind, it is possible that studies with relevant information were overlooked as their methods may have been too dissimilar to the chosen inclusion criteria used for this systematic literature review. Although the gender breakdown of participants in this review is similar (*n* = 50 male, *n* = 55 female), this is surprising as other authors research suggests that the strong gender gap in this sport nutrition area remains [90,91]. Sample sizes were generally small, with the number of study participants ranging from 1 to 25, with a mean of approximately *n* = 15 participants. The variation in protocols and GIS rating systems made it difficult to meta-analyze the results. The assorted scaling systems utilized to quantify GI discomfort, namely VAS, Likert-type, and modified BORG scales, made it difficult to compare the success of the various nutrition interventions and created disparities between the validity and reproducibility of studies [54]. Moreover, each study design possesses its own limitations. For example, where case studies offer detailed insight, their findings lack generalizability. Laboratory-based trials allow for the control of confounders but fail to accurately mimic real-world scenarios and the efficacy of interventions [92]. It is important that these are considered when extracting and employing the research findings. The majority of studies suggest that larger sample sizes are needed to overcome individual intolerances and differences in responses to interventions, which may potentially distort results. In addition to this, of the included seven studies, three were conducted under laboratory conditions, with their efficacy in real-life situations yet to be investigated. Of those that were also conducted outside of the laboratory, the participants were often supplied with their meals, and their adherence was monitored with the use of diet and training logs. Caution must be taken when applying such findings in a real-world scenario.

## 6. Conclusions

The findings of this systematic literature review suggest that following a diet low in FODMAPs has the potential to support ultra-endurance athletes in the management of their exercise-induced GIS leading up to and during exercise, as well as throughout the recovery period. Carbohydrates may be more favorable as an energy source during ultra-endurance exercise training and racing, with their adaptation through gut training also existing as a promising strategy to further improve their tolerability. However, research findings regarding the precise carbohydrate blend for optimum performance effects with minimum GIS incidence remain trivial and inconclusive. The amount of lactose present in pre-exercise meals and those during the recovery period, similar to the findings for that of MCT supplementation, does not appear to be beneficial for exercise-associated GIS attenuation. The findings of this review may be used to guide future research and support sports professionals in their recommendations of practices that improve athletic performance by attenuating such gastrointestinal symptoms. Further research should recruit larger sample sizes, supported by a power calculation, and focus on investigating the impact of specific foods and nutrients on exercise-induced GIS complaints and their associated mechanisms.

## Figures and Tables

**Figure 1 nutrients-15-04330-f001:**
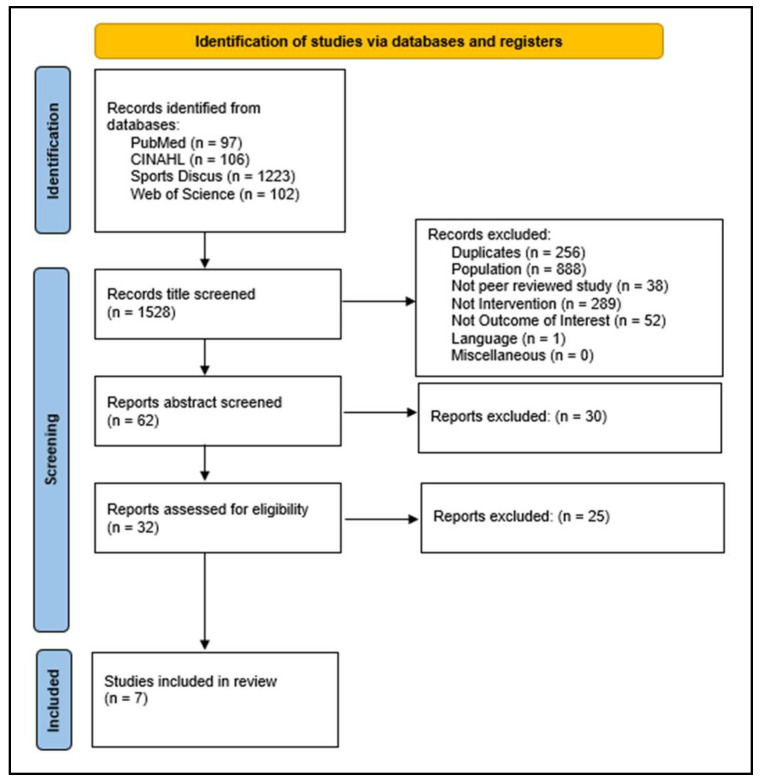
Prisma flow diagram showing the steps of the search strategy for this systematic literature review, resulting in seven studies for inclusion in this systematic literature review.

**Table 1 nutrients-15-04330-t001:** The PICOS model for the definition of inclusion criteria in this systematic literature review.

	Topic	Criteria
P	Population	Humans ≥18 years old are considered “ultra-endurance athletes” by the research authors.
I	Intervention	Prescribed nutritional intervention to alter gastrointestinal symptoms.
C	Comparators	None/placebo.
O	Outcomes	Altered gastrointestinal health (prevalence, duration, and severity of GI symptoms).
S	Study design	Prospective randomized controlled trials, crossover trials, case studies.

**Table 2 nutrients-15-04330-t002:** Search concepts and their related terms searched for conducting this systematic literature review, with the asterisk serving as a truncation operator.

**Concept 1:** “ultra-endurance” OR “ultra-athlete” OR ironman OR “ultra-endurance training” OR “ultra-distance” OR “ultramarathon” OR “ultra-event”
**Concept 2:** “nutritional intervention” OR nutrition* OR food OR diet OR diets OR “dietary pattern” OR carbohydrate* OR fat OR fats OR “fatty acids” OR “dietary fats” OR protein* OR antioxidant* OR supplement* OR fasting OR hydration OR drink* OR beverage* OR energy OR macronutrient* OR micronutrient* OR keto* OR glucose OR sugar OR calorie* OR prebiotic OR probiotic
**Concept 3:** gastrointestinal OR “gastrointestinal problems” OR “gastrointestinal symptoms” OR “gastrointestinal events” OR “GI” OR “GI problems” OR “GI symptoms” OR “GI events” OR vomiting OR diarrhea OR constipation OR nausea OR “abdominal pain” OR pain OR discomfort OR ache OR stitch OR bloating OR reflux OR cramp* OR abdominal OR digest* OR stomach OR intestinal OR gut OR “gut microbiome” OR microbiota OR “nutritional manipulation”

**Table 3 nutrients-15-04330-t003:** Overview of the subjects, supplementation duration, dietary conditions, gastrointestinal distress recordings, and main findings of the studies included in this review, where GI = gastrointestinal, GIS = gastrointestinal symptoms, MCFA = medium-chain-fatty-acids, LCFA = long-chain-fatty-acids, VAS = Visual Analogue Scale, CHO = carbohydrate(s), FODMAP = fermentable oligo-, di-, and monosaccharides and polyols.

Trial (First Author)	Year	*n*	Sex	Characteristics	Setting/Condition	Duration of Supplement Use/Dietary Intervention in Days	Diet Labels	Energy	CHO	Protein	Fat	Other	GIS Recordings	Main Findings	ADA Quality Assessment
Goedecke	2005	9	M	Competitive ultra-cyclists.	Laboratory	2 (1 day with each supplement).	Medium-chain triacylglycerol (MCT)	-	-	-	-	Pre: 32 g MCT, In: 200 mL 4.3% MCT, 10% CHO solution every 20 m.	0–3 scale.	50% of participants suffered GIS during or after MCT trial (*n* = 1 mild and *n* = 3 severe). No GIS reported during control trial.	+
							Control	-	Pre: 75 g, In: 200 mL 10% CHO solution every 20 m	-	-	-			
Gaskell	2019	1	F	Recreational ultrarunner. IBS diagnosed.	Field	7	Pre-trial	-	-	-	-	FODMAPs: 3.9 g.	100 mm VAS at rest and during training.	Successfully implemented low-FODMAP diet. Severe GIS during training but modest GIS (bloating and flatulence) during race. Severe nausea (potentially due to low energy intake).	+
							During trial	-	-	-	-	FODMAPS (excluding during racing): 5.1 g.			
							Placebo (Water)	-	-	-	-	Water.			
Gaskell	2020	18	MF	Endurance and ultra-endurance runners.	Laboratory	1 day on each diet.	High FODMAP	2645 ± 747 g	405 ± 124 g	102 ± 29 g	68 ± 18 g	FODMAPs: 46.9 ± 26.2 g.	10-point modified VAS every 15 mins.	GIS severity (*p* = 0.056), total (*p* = 0.014), upper (*p* = 0.019), and lower (*p* = 0.006) were higher as a whole in response to the high-FODMAP diet.	+
							Low FODMAP	2375 ± 538 g	355 ± 89 g	91 ± 23 g	66 ± 15 g	FODMAPs: 2.0 ± 0.7 g.			
Haakonssen	2014	25	F	Competitive cyclists.	Laboratory	2 (1 day with each supplement).	Dairy	54 ± 2 kJ·kg^−^¹	2 ± 0 g·kg^−^¹	0.6 ± 0.1 g·kg^−^¹	0.3 ± 0.0 g·kg^−^¹	Pre-exercise meal.	100 mm VAS; 5 different timepoints.	No significantly significant association between pre-trial gut discomfort and meal type (*p* = 0.15). No statistically significant association between gut comfort delta scores and meal type at 30 min (*p* = 0.31) or 60 min (*p* = 0.17) post-meal. Dairy meal may be more palatable.	+
							Control	54 ± 2 kJ·kg^−^¹	2 ± 0 g·kg^−^¹	0.2 ± 0.0 g·kg^−^¹	0.4 ± 0.0 g·kg^−^¹	Pre-exercise meal.			
							High-CHO, low-frequency	-	2.4 g/min	-	-	5 min pre-race, every 15 km in race.			
							Moderate-CHO, high frequency	-	1.2 g/min	-	-	5 min pre-race, every 5 km in race.			
							Moderate-CHO, low frequency	-	1.2 g/min	-	-	5 min pre-race, every 15 km in race.			
Russo	2021	9	MF	Recreationally and competitively trained endurance and ultra-endurance athletes.	Field	8	Low FODMAP (24 h pre- and throughout trial)	-	364 g	101 g	32 g	FODMAPS: <2 g per meal. Those with greater BMed were provided with additional meal servings/snacks.	Modified VAS. On day 1: pre-trials, during the final 30 s of cycling, and every 30 min during recovery period. On day 2: on arrival and after performance test.	Greater total gut discomfort on MBSB (*p* = 0.053). No significant effects or interactions observed for upper-GIS, nausea, or total-GIS.	+
							Chocolate-flavored dairy milk [54]	2715 kJ	92 g	30 g	17 g	Served in 3 equal boluses every 10 min, beginning 30 min into recovery.			
							Chocolate-flavored dairy milk-based supplement drink (MBSB)	4029 kJ	170 g	63 g	2 g				
Costa	2017	25	MF	10F and 15M. Recreationally competitive endurance and ultra-endurance runners.	Field	14 (2 weeks in 1 of 3 protocols).	CHO-gel disk (CHO-S)	-	30 g	-	-	While running for 10 days over 2 weeks.	10-point Likert-type rating scale.	No significant differences in GIS between groups in trial 1 and 2. Significant improvements in gut discomfort (*p* = 0.018), total GIS (*p* = 0.003), upper GIS (*p* = 0.043), lower GIS (*p* = 0.010), and nausea (*p* = 0.050) observed in CHO-S and CHO-F compared with placebo.	+
							CHO-food (CHO-F)	-	30 g	-	-	While running for 10 days over 2 weeks.			
							Placebo	-	0 g	-	-	While running for 10 days over 2 weeks.			
Miall	2017	18	MF	10M and 8FM. Recreationally competitive endurance and ultra-endurance runners.	Field	10 (5 with supplement, 2 resting without supplement, 5 more with supplement, 2 more resting without supplement).	CHO-gel disc (CHO)	-	30 g	-	-	At 0, 20, and 40 min.	10-point Likert-type rating scale for feeding tolerance. Every 10 mins during gut trials.	All participants reported at least one GIS during pre-intervention trial. 67% reported at least one of these as severe. GIS more common in CHO (80%) than PLA (50%). Subjects accustomed to CHO during training reported significantly reduced gut discomfort (*p* = 0.001), total (*p* = 0.002), upper (*p* = 0.001) GIS, and nausea (*p* = 0.026). Significant reduction in gut discomfort on CHO (*p* < 0.001). No gut discomfort improvements observed on PLA.	+
							Placebo (PLA)	-	0 g	-	-	At 0, 20, and 40 min.			
							Gel	-	-	-	-	Glucose/fructose in 2:1 ratio.			
							Drink mix powder	-	-	-	-	Glucose/fructose in 2:1 ratio.			
							Control	-	-	-	-	Glucose/maltodextrin only.			

## Data Availability

The data extracted for analysis and use in this review can be made available upon request.

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
