# Peer review of "Exploring the Nutrition Strategies Employed by Ultra-Endurance Athletes to Alleviate Exercise-Induced Gastrointestinal Symptoms—A Systematic Review"

_nutrients, 2023, doi:10.3390/nu15204330_

Round 1

Reviewer 1 Report

A very well written and executed review of an important topic in ultra-endurance sports/exercise. GIS are extremely common in sports such as ultra running events, ironman triathlon (or longer Ultra-man events), ultra-distance cycling events and adventure racing.  This reviewer has had direct experience with symptoms. One may see GIS feature as a significant topic of comment in-pre/mid/post race interviews featuring top and recreational ultra distance athletes (e.g.  viewing of several U Tube videos of this year's UTMB [with 3000+ participants] and other large ultra trail running or cycling events).  With the explosion of participation in these events, nutritional and fueling strategies and mitigation of GIS's emerge as the most important issues arising for athletes during competition especially in challenging environmental conditions ( equal to orthopedic injury, sleep deficit, hypoxia, skin irritations etc.).  Given the popularity of low CHO diets ("Keto" diets), train low (fat) and compete high (CHO), and the concept of CHO gut training to increase intake sometime to > 110 g/hr, the topic is of interest. However the fact that after this significant literature search, only a few well-powered studies were found, illustrates the gap in knowledge remaining on this topic.

Generally I find the manuscript is well crafted and organized with nice flow. I would suggest creating a few more paragraph breaks for reader ease.  If it is permitted by the journal, a list of acronyms after the abstract would be helpful.  The limitation and conclusion sections are well written and summarized nicely.  

Specific comments:

1. Pg 2 lines 85-87;   regarding the influence of environmental factors: temperature and humidity are mentioned, but perhaps altitude (mild hypoxia) could be elaborated upon further, as many of the extreme trail running events include large segments above 1500m which combine large elevation gains (with associated increases in muscle CHO metabolic demands).  Extreme endurance events at high altitude may pose additional challenges to the gut due to increased dehydration, changes in gut acid base balance and muted nutritional and gut tolerance .  Also extreme cold also hinders nutritional strategies and my exacerbate GIS during prolonged events (such as ultra ski events, trans-polar expeditions and prolonged military Arctic operations).

2.  Pg. 6  Table 2.   Suggest re-format in Landscape mode, in order to allow the text to wrap and maintain readability/continuity of the data.  One variable that may have been of interest to include in the Table would be exercise /trial/ competition duration (especially in the field studies), as the range can be wide: from a couple of hours in the case of laboratory studies, all the way up to 20+ hours  or several days, for ultra endurance events.

Author Response

Reviewer 1

A very well written and executed review of an important topic in ultra-endurance sports/exercise. GIS are extremely common in sports such as ultra running events, ironman triathlon (or longer Ultra-man events), ultra-distance cycling events and adventure racing.  This reviewer has had direct experience with symptoms. One may see GIS feature as a significant topic of comment in-pre/mid/post race interviews featuring top and recreational ultra distance athletes (e.g.  viewing of several U Tube videos of this year's UTMB [with 3000+ participants] and other large ultra trail running or cycling events).  With the explosion of participation in these events, nutritional and fueling strategies and mitigation of GIS's emerge as the most important issues arising for athletes during competition especially in challenging environmental conditions ( equal to orthopedic injury, sleep deficit, hypoxia, skin irritations etc.).  Given the popularity of low CHO diets ("Keto" diets), train low (fat) and compete high (CHO), and the concept of CHO gut training to increase intake sometime to > 110 g/hr, the topic is of interest. However the fact that after this significant literature search, only a few well-powered studies were found, illustrates the gap in knowledge remaining on this topic.

Generally I find the manuscript is well crafted and organized with nice flow. I would suggest creating a few more paragraph breaks for reader ease.  If it is permitted by the journal, a list of acronyms after the abstract would be helpful.  The limitation and conclusion sections are well written and summarized nicely.  

Response: Thank you. The authors are extremely grateful for the time you’ve taken to review this work and all of the feedback you have provided. Regarding abbreviations, the journal guidelines recommend explaining the abbreviation on its first use in the paper.

Specific comments:

1. Pg 2 lines 85-87;  regarding the influence of environmental factors: temperature and humidity are mentioned, but perhaps altitude (mild hypoxia) could be elaborated upon further, as many of the extreme trail running events include large segments above 1500m which combine large elevation gains (with associated increases in muscle CHO metabolic demands).  Extreme endurance events at high altitude may pose additional challenges to the gut due to increased dehydration, changes in gut acid base balance and muted nutritional and gut tolerance .  Also extreme cold also hinders nutritional strategies and my exacerbate GIS during prolonged events (such as ultra ski events, trans-polar expeditions and prolonged military Arctic operations).

Response: The introduction has now been re-structured to better connect related topics. 70-71 now explain how environmental conditions, including altitude, can have effects on GIS.

2.  Pg. 6  Table 2.   Suggest re-format in Landscape mode, in order to allow the text to wrap and maintain readability/continuity of the data.  One variable that may have been of interest to include in the Table would be exercise /trial/ competition duration (especially in the field studies), as the range can be wide: from a couple of hours in the case of laboratory studies, all the way up to 20+ hours  or several days, for ultra endurance events.

Response: The table has been re-formatted to Landscape and information regarding the trial durations has been added. This has been submitted in a separate document.

Reviewer 2 Report

Abstract

·         The background is lengthy. While it is crucial to set context, consider being more concise and to the point.

·         It is mentioned that "Data investigating such practices is sparse," yet the aim of the research does not fully resonate with this statement. The statement of the problem and the aim of the research must align perfectly.

·         Good detail on the databases used. However, consider including the search terms or keywords employed during the search to give the reader an idea about the specificity of the search.

·         Good mention of the design of the studies included. However, consider providing more clarity about the participants' duration, region, or demographic.

·         Instead of presenting the non-significant findings (e.g., lactose content), emphasize significant findings and their implications more.

·         While it is appreciated that the limitations are pointed out, the conclusion section should also emphasize the main takeaways from the review.

·         The statement "should focus on the impact of specific foods and nutrients on such symptoms, as well as their associated mechanisms" seems more like a future recommendation than a conclusion.

·         Be cautious of redundant information. For example, the potential benefits of a low FODMAP diet are mentioned twice in the abstract.

·         Some terms, such as "FODMAPs", could benefit from a brief description or example when first introduced to aid readers unfamiliar with the term.

·         The abstract gives a broad overview of the systematic review, but some elements could be streamlined, and certain key details could be added for a more comprehensive understanding. The grammar and flow of the abstract are generally good but could benefit from a slight refinement.

Introduction

·         The introduction is detailed and provides a thorough background on the topic. However, the structure could benefit from streamlining. It covers multiple aspects, from endurance sports to nutritional interventions and gastrointestinal symptoms (GIS) to IBS. Grouping related information more tightly and eliminating redundant details would improve readability.

·         The definition of ultra-endurance activity ranges from exceeding four hours to at least six hours. While it is essential to acknowledge varying definitions, the authors should specify which definition they are adopting for this review for clarity.

·         Some information appears to be repetitive. For example, the effect of extreme environmental conditions on GIS is mentioned multiple times. Consider consolidating similar points to prevent redundancy.

·         It is commendable to have references supporting the claims. However, ensure that references are appropriately cited and consistent and that the most relevant ones are chosen.

·         The segment discussing IBS is detailed but might benefit from a more straightforward presentation. Consider simplifying descriptions and focusing on the most crucial aspects of the study's aims.

·         While terms like "GIS" and "IBS" are defined, ensuring that all technical terms are sufficiently defined upon first mention is vital to cater to readers less familiar with the specific topic.

The introduction does a commendable job discussing GIS in endurance athletes. However, there could be more emphasis on the link between nutrition and GIS in this population to better set the stage for the review's aim.

·         The transition from discussing IBS and GIS in athletes to the specific aim of the systematic review can be smoother. Consider using transition sentences that guide the reader naturally to the study's primary purpose.

·         The aim of the systematic review is clear at the end. However, this could be emphasized more, possibly with its subsection or highlighted in some way to stand out.

·         The introduction is somewhat lengthy. While a detailed introduction is essential for setting the context, it is crucial to strike a balance to ensure readers can quickly grasp the main points without feeling overwhelmed.

·         In discussing various nutritional interventions, the authors might consider specifying whether these interventions are evidence-based or anecdotal. This could set the stage for the necessity of a systematic review.

·         The introduction provides a comprehensive background on ultra-endurance sports, nutrition, and gastrointestinal issues. However, refining its structure, removing redundancies, and ensuring a smoother flow will significantly improve its clarity and impact. The clear aim statement at the end is well done and directs the review.

Methods

·         It is good to see that multiple databases were utilized for the literature search. This helps ensure a comprehensive review.

·         The authors updated their search in January 2023, ensuring that their review remains current and includes the latest relevant publications.

·         While the section states that keywords were defined, it does not explicitly list all keywords and combinations used. Listing specific search terms can improve the reproducibility of the search.

·         The inclusion and exclusion criteria are clearly outlined, strengthening the review.

·         It might be helpful to define what “GB” stands for earlier in the section (it's inferred to mean gastrointestinal symptoms or GIS, but clarity is important).

·         The exclusion criterion of studies "not freely available in the English language" might limit the comprehensiveness of the review. While language restrictions are common, deciding to exclude based on availability may introduce bias.

·         It is good practice to have multiple reviewers (LR and ED), which enhances the validity of the study selection process.

·         It is helpful to see that the authors used standardized software for data organization (Microsoft Excel). However, this section might benefit from details on how the extracted data will be synthesized and analyzed or if any statistical methods will be applied.

·         It is commendable that a recognized tool (Academy of Nutrition and Dietetics Quality Criteria Checklist) was used to assess study quality; however, it may be beneficial to describe briefly why this particular tool was chosen over others and its relevance to this specific topic.

Minor phrasing concerns

·         "Scientific literature was systemically searched" should be "systematically searched".

·         In the section "A systemic literature search was conducted", the word should be "systematic", not "systemic".

Power calculations:

·         It is good that the authors highlighted power calculations in some of the studies. This is an essential aspect of study quality and can affect the interpretation of results.

Results:

·         The article screening and selection process was conducted by a single author (TR), which might introduce bias. Best practice suggests that at least two reviewers independently screen articles for inclusion to minimize errors and bias.

·         There are inconsistent descriptions regarding rating scales for GIS. The text should clarify how each scale correlates with the severity of GIS.

Study-specific critiques:

·         Low FODMAP Gaskell [50]: A single case study with inherent generalizability limitations. However, it provides valuable insights, especially for niche populations.

·         Haakonssen et al. [53]: The lack of full blinding due to meal differences could introduce bias. Also, there was no significant association reported, which could suggest that lactose might not play a significant role in GIS for the population studied.

·         Russo et al. [54]: The small sample size, especially the unequal male-to-female ratio, could affect the generalizability of the findings.

·         Goededce et al. [55]: The study highlights potential issues with MCTs when combined with carbs, but the sample size is small. It would be helpful to know if participants had prior exposure to MCTs as they can cause GIS in individuals unaccustomed to them.

·         Costa et al. [56]: Introducing a parallel-group trial is interesting, but the results seem truncated and do not provide clear outcomes or implications.

Consistency and clarity:

·         The section sometimes reads like a mixture of results and methods. Separating the methodological aspects (like blinding processes) from the actual results would enhance clarity.

General comments:

·         Given the variety of studies, methods, and interventions reviewed, it might be beneficial to introduce a more structured approach to presenting each study. For instance, consistently outlining the study design, participant characteristics, interventions, outcomes, and significance levels would provide better clarity to readers.

·         The text has occasional grammar and typographical errors, which should be addressed for clarity and professionalism.

Discussion:

·         The “Discussion” is dense, detailed, and slightly repetitive in places. For easier comprehension, it may benefit from subheadings separating the various content sections, such as FODMAPs, Lactose-rich pre-exercise meals, MCTs, and so on.

·         Some themes and findings, such as the benefits of a low FODMAP diet for GIS management, were reiterated multiple times. A consolidation of ideas could make the section more concise.

·         There is a slight inconsistency in referencing the studies. For example, Gaskell and Costa are referred to using both their last names and only by Gaskell in different parts of the text. Consistency is key to avoid confusion.

·         While various studies (case studies, controlled trials, etc.) are included, there is minimal discussion on the limitations inherent to each study type. For instance, case studies offer detailed insights but cannot be generalized. Highlighting the limitations ensures a well-rounded discussion.

·         Some studies present conflicting results. While this is addressed, there is a lack of deep dive into possible reasons for these discrepancies. Discussing potential variables that could account for differing results will enrich the review.

·         While numerous studies are cited, there is a limited critique of the methodologies employed. An expert review could delve deeper into sample size, control for confounding variables, and other study design components that might influence outcomes.

·         The “Discussion” frequently mentions p-values to indicate significance but lacks a clear distinction between statistical and clinical significance. For some findings, even if

·         The discussion on lactose-rich meals touches upon potential nutritional deficiencies when removing dairy from the diet. However, it does not consider the increasing prevalence of lactose intolerance in some populations nor acknowledge alternative sources of essential nutrients.

·         While the discussion gives a comprehensive overview of the existing literature, it could benefit from a stronger emphasis on concrete recommendations based on the findings and clear directions for future research.

·         In summary, while the discussion provides a thorough overview of various studies related to diet and gastrointestinal issues during endurance sports, it would benefit from improved organization, more in-depth critique, clearer recommendations, and a concise summary.

Some editing is required to enhance clarity, reduce repetitiveness, and correct minor typographical errors. Some sections might benefit from restructuring for better flow and comprehension.

Author Response

Reviewer 2

Response: The authors thank you for all of your feedback and the time dedicated to reviewing this systematic literature review.

Abstract

  • The background is lengthy. While it is crucial to set context, consider being more concise and to the point.

Response: Background has been reduced.

  • It is mentioned that "Data investigating such practices is sparse," yet the aim of the research does not fully resonate with this statement. The statement of the problem and the aim of the research must align perfectly.

 Response: This statement has been re-phrased to better align with the aim of this research.

  • Good detail on the databases used. However, consider including the search terms or keywords employed during the search to give the reader an idea about the specificity of the search.

Response: Key words are listed under the abstract but have now also been mentioned in the methods section of the abstract.

  • Good mention of the design of the studies included. However, consider providing more clarity about the participants' duration, region, or demographic.

Response: Participant gender breakdown has been added.

  • Instead of presenting the non-significant findings (e.g., lactose content), emphasize significant findings and their implications more.

Response: This has been removed.

  • While it is appreciated that the limitations are pointed out, the conclusion section should also emphasize the main takeaways from the review.

Response: This has been re-worded. Limitations are now not mentioned first, but after the main conclusion.

  • The statement "should focus on the impact of specific foods and nutrients on such symptoms, as well as their associated mechanisms" seems more like a future recommendation than a conclusion.

Response: This statement has been removed from the abstract.

  • Be cautious of redundant information. For example, the potential benefits of a low FODMAP diet are mentioned twice in the abstract.

Response: This has been re-worded in the conclusion.

  • Some terms, such as "FODMAPs", could benefit from a brief description or example when first introduced to aid readers unfamiliar with the term.

Response: A brief description of “short-chain, poorly absorbed carbohydrates” has been added.

  • The abstract gives a broad overview of the systematic review, but some elements could be streamlined, and certain key details could be added for a more comprehensive understanding. The grammar and flow of the abstract are generally good but could benefit from a slight refinement.

Response: Your recommended changes have now been made so that the abstract contains more key details and aids the reader in fully understanding the research.

Introduction

  • The introduction is detailed and provides a thorough background on the topic. However, the structure could benefit from streamlining. It covers multiple aspects, from endurance sports to nutritional interventions and gastrointestinal symptoms (GIS) to IBS. Grouping related information more tightly and eliminating redundant details would improve readability.

Response: This has been re-organised with related information more closely grouped.

  • The definition of ultra-endurance activity ranges from exceeding four hours to at least six hours. While it is essential to acknowledge varying definitions, the authors should specify which definition they are adopting for this review for clarity.

Response: This has been clarified in the final line of the introduction, and in the inclusion criteria of the methods section.

  • Some information appears to be repetitive. For example, the effect of extreme environmental conditions on GIS is mentioned multiple times. Consider consolidating similar points to prevent redundancy.

Response: This repetition has been corrected and repeating phrases have been removed.

  • It is commendable to have references supporting the claims. However, ensure that references are appropriately cited and consistent and that the most relevant ones are chosen. Recent references have been incorporated, for example reference 20, available at https://pubmed.ncbi.nlm.nih.gov/35043679/
  • The segment discussing IBS is detailed but might benefit from a more straightforward presentation. Consider simplifying descriptions and focusing on the most crucial aspects of the study's aims.

Response: Previously, less information on IBS was included but the authors are aware that not all readers will have knowledge on IBS.

  • While terms like "GIS" and "IBS" are defined, ensuring that all technical terms are sufficiently defined upon first mention is vital to cater to readers less familiar with the specific topic.

The introduction does a commendable job discussing GIS in endurance athletes. However, there could be more emphasis on the link between nutrition and GIS in this population to better set the stage for the review's aim.

Response: The introduction has now been restructured so that, for example, GIS is explained at the beginning of the paragraph after it is first mentioned (line 64).

  • The transition from discussing IBS and GIS in athletes to the specific aim of the systematic review can be smoother. Consider using transition sentences that guide the reader naturally to the study's primary purpose.

Response: The intro is now re-structured so that related areas are more smoothly linked.

  • The aim of the systematic review is clear at the end. However, this could be emphasized more, possibly with its subsection or highlighted in some way to stand out. Response: The aim now sits in a separate section to highlight it.
  • The introduction is somewhat lengthy. While a detailed introduction is essential for setting the context, it is crucial to strike a balance to ensure readers can quickly grasp the main points without feeling overwhelmed.

Response: The introduction has been re-structured so that it is easier for readers to follow and sentences have been shortened.

  • In discussing various nutritional interventions, the authors might consider specifying whether these interventions are evidence-based or anecdotal. This could set the stage for the necessity of a systematic review.

Response: This has been incorporated (eg line 129).

  • The introduction provides a comprehensive background on ultra-endurance sports, nutrition, and gastrointestinal issues. However, refining its structure, removing redundancies, and ensuring a smoother flow will significantly improve its clarity and impact. The clear aim statement at the end is well done and directs the review.

Response: Your recommendations have been made. The introduction has also been restructured to better group together related topics.

Methods

  • It is good to see that multiple databases were utilized for the literature search. This helps ensure a comprehensive review.
  • The authors updated their search in January 2023, ensuring that their review remains current and includes the latest relevant publications.
  • While the section states that keywords were defined, it does not explicitly list all keywords and combinations used. Listing specific search terms can improve the reproducibility of the search.

Response: The keywords have now been included in this section as Table 2.

  • The inclusion and exclusion criteria are clearly outlined, strengthening the review.
  • It might be helpful to define what “GB” stands for earlier in the section (it's inferred to mean gastrointestinal symptoms or GIS, but clarity is important).

Response: After searching for “GB” it is not in the text.

  • The exclusion criterion of studies "not freely available in the English language" might limit the comprehensiveness of the review. While language restrictions are common, deciding to exclude based on availability may introduce bias.

Response: this has been added to the limitations section

  • It is good practice to have multiple reviewers (LR and ED), which enhances the validity of the study selection process.
  • It is helpful to see that the authors used standardized software for data organization (Microsoft Excel). However, this section might benefit from details on how the extracted data will be synthesized and analyzed or if any statistical methods will be applied.

Response: Information explaining that a statistical analysis has not been conducted due to study heterogeneity has now been added.

  • It is commendable that a recognized tool (Academy of Nutrition and Dietetics Quality Criteria Checklist) was used to assess study quality; however, it may be beneficial to describe briefly why this particular tool was chosen over others and its relevance to this specific topic.

Response: Reasoning for using this tool has now been included.

Minor phrasing concerns

  • "Scientific literature was systemically searched" should be "systematically searched".

Response: This has been corrected.

  • In the section "A systemic literature search was conducted", the word should be "systematic", not "systemic".

Response: This has been corrected.

Power calculations:

  • It is good that the authors highlighted power calculations in some of the studies. This is an essential aspect of study quality and can affect the interpretation of results.

Results:

  • The article screening and selection process was conducted by a single author (TR), which might introduce bias. Best practice suggests that at least two reviewers independently screen articles for inclusion to minimize errors and bias.

Response: the first author was supervised by LR and ED and at each stage the results were independently reviewed by LR and ED and the group met to discuss and resolve any conflict.

  • There are inconsistent descriptions regarding rating scales for GIS. The text should clarify how each scale correlates with the severity of GIS.

Response: This has been explained on line 269 now

Study-specific critiques:

  • Low FODMAP Gaskell [50]: A single case study with inherent generalizability limitations. However, it provides valuable insights, especially for niche populations.

Response: This has been added

  • Haakonssen et al. [53]: The lack of full blinding due to meal differences could introduce bias. Also, there was no significant association reported, which could suggest that lactose might not play a significant role in GIS for the population studied.

Response: This has been added

  • Russo et al. [54]: The small sample size, especially the unequal male-to-female ratio, could affect the generalizability of the findings.

Response: This has been added

  • Goededce et al. [55]: The study highlights potential issues with MCTs when combined with carbs, but the sample size is small. It would be helpful to know if participants had prior exposure to MCTs as they can cause GIS in individuals unaccustomed to them.

Response: This information has been added

  • Costa et al. [56]: Introducing a parallel-group trial is interesting, but the results seem truncated and do not provide clear outcomes or implications.

Response: This information has been added

Consistency and clarity:

  • The section sometimes reads like a mixture of results and methods. Separating the methodological aspects (like blinding processes) from the actual results would enhance clarity. Response: The methodology of each study has now been moved to the top of each paragraph so they all flow the same.

General comments:

  • Given the variety of studies, methods, and interventions reviewed, it might be beneficial to introduce a more structured approach to presenting each study. For instance, consistently outlining the study design, participant characteristics, interventions, outcomes, and significance levels would provide better clarity to readers.

Response: The accompanying table has been edited and changed to landscape layout with this information so it is easier to read and follow.

  • The text has occasional grammar and typographical errors, which should be addressed for clarity and professionalism.

Response: This has now been proof-read and errors corrected.

Discussion:

  • The “Discussion” is dense, detailed, and slightly repetitive in places. For easier comprehension, it may benefit from subheadings separating the various content sections, such as FODMAPs, Lactose-rich pre-exercise meals, MCTs, and so on.

Response: the discussion is separated by subheadings, but these have now been changed to italics so that they are more obvious for readers.

  • Some themes and findings, such as the benefits of a low FODMAP diet for GIS management, were reiterated multiple times. A consolidation of ideas could make the section more concise. Response: The discussion has been reviewed and repetitions have been removed.
  • There is a slight inconsistency in referencing the studies. For example, Gaskell and Costa are referred to using both their last names and only by Gaskell in different parts of the text. Consistency is key to avoid confusion.

Response: This has been corrected, Gaskell and Costa is reference 50 and Gaskell et al is reference 51

  • While various studies (case studies, controlled trials, etc.) are included, there is minimal discussion on the limitations inherent to each study type. For instance, case studies offer detailed insights but cannot be generalized. Highlighting the limitations ensures a well-rounded discussion. Response: This has been added to the limitations section
  • Some studies present conflicting results. While this is addressed, there is a lack of deep dive into possible reasons for these discrepancies. Discussing potential variables that could account for differing results will enrich the review.

Response: This has now been included, noting that [51] did not incorporate a low FODMAP diet which could affect GIS.

  • While numerous studies are cited, there is a limited critique of the methodologies employed. An expert review could delve deeper into sample size, control for confounding variables, and other study design components that might influence outcomes.

Response: This information is in the limitations section.

  • The “Discussion” frequently mentions p-values to indicate significance but lacks a clear distinction between statistical and clinical significance. For some findings, even if

Response: incomplete comment, but the statistical significance recognised at <0.05 has been added to line 239.

  • The discussion on lactose-rich meals touches upon potential nutritional deficiencies when removing dairy from the diet. However, it does not consider the increasing prevalence of lactose intolerance in some populations nor acknowledge alternative sources of essential nutrients.

Response: Information on the prevalence of lactose intolerance has now been added.

  • While the discussion gives a comprehensive overview of the existing literature, it could benefit from a stronger emphasis on concrete recommendations based on the findings and clear directions for future research.

Response: All of the recommended changes have been made, and sentences have been restructured/added to solidify the findings and recommendations for future research.

  • In summary, while the discussion provides a thorough overview of various studies related to diet and gastrointestinal issues during endurance sports, it would benefit from improved organization, more in-depth critique, clearer recommendations, and a concise summary.

Comments on the Quality of English Language

Some editing is required to enhance clarity, reduce repetitiveness, and correct minor typographical errors. Some sections might benefit from restructuring for better flow and comprehension.

Response: The document has now been reviewed and updated.